**Peer**J

# Expression profiles of the immune genes CD4, CD8$\beta$, IFN$\gamma$, IL-4, IL-6 and IL-10 in mitogen-stimulated koala lymphocytes (*Phascolarctos cinereus*) by qRT-PCR

Iona E. Maher, Joanna E. Griffith, Quintin Lau, Thomas Reeves and Damien P. Higgins

Faculty of Veterinary Science, The University of Sydney, NSW, Australia

## ABSTRACT

Investigation of the immune response of the koala (*Phascolarctos cinereus*) is needed urgently, but has been limited by scarcity of species-specific reagents and methods for this unique and divergent marsupial. Infectious disease is an important threat to wild populations of koalas; the most widespread and important of these is Chlamydial disease, caused by *Chlamydia pecorum* and *Chlamydia pneumoniae*. In addition, koala retrovirus (KoRV), which is of 100% prevalence in northern Australia, has been proposed as an important agent of immune suppression that could explain the koala's susceptibility to disease. The correct balance of T regulatory, T helper 1 (Th1) and Th2 lymphocyte responses are important to an individual's susceptibility or resistance to chlamydial infection. The ability to study chlamydial or KoRV pathogenesis, effects of environmental stressors on immunity, and the response of koalas to vaccines under development, by examining the koala's adaptive response to natural infection or in-vitro stimulation, has been limited to date by a paucity of species- specific reagents. In this study we have used cytokine sequences from four marsupial genomes to identify mRNA sequences for key T regulatory, Th1 and Th2 cytokines interleukin 4 (IL-4), interleukin 6 (IL-6), interleukin 10 (IL-10) and interferon gamma (IFN$\gamma$ ) along with CD4 and CD8$\beta$. The koala sequences used for primer design showed $>58\%$ homology with grey short-tailed opossum, $>71\%$ with tammar wallaby and 78% with Tasmanian devil amino acid sequences. We report the development of real-time RT-PCR assays to measure the expression of these genes in unstimulated cells and after three common mitogen stimulation protocols (phorbol myristate acetate/ionomycin, phorbol myristate acetate/phytohemagglutinin and concanavalin A). Phorbol myristate acetate/ionomycin was found to be the most effective mitogen to up-regulate the production of IL-4, IL-10 and IFN$\gamma$ . IL-6 production was not consistently up-regulated by any of the protocols. Expression of CD4 and CD8$\beta$ was down-regulated by mitogen stimulation. We found that the reference genes GAPDH and 28s are valid for normalising cytokine expression by koala lymphocytes after mitogen stimulation.

Corresponding author
Damien P. Higgins,
damien.higgins@sydney.edu.au

## INTRODUCTION

There is a long-standing and growing need to understand the immune system of the koala (*Phascolarctos cinereus*). Chlamydial disease, caused by *Chlamydia pecorum* and, to a lesser extent, *Chlamydia pneumoniae*, manifests as keratoconjunctivitis, cystitis, pyelonephritis and reproductive tract inflammation and fibrosis, leading to infertility (*Hemsley & Canfield, 1997*; *Higgins, Hemsley & Canfield, 2005*; *Obendorf, 1983*). It is a common cause of morbidity and mortality (*Polkinghorne, Hanger & Timms, 2013*) and, in combination with habitat loss and population fragmentation, bush fires, drought and heat waves, motor vehicle accidents and dog attacks (*Dique et al., 2004*, *2003*; *Griffith et al., 2013*; *Lunney et al., 2012*, *2007*; *Melzer et al., 2000*; *Obendorf, 1983*; *Reed & Lunney, 1990*; *Rhodes et al., 2011*; *Seabrook et al., 2011*) it has contributed to the koala being listed as a vulnerable species over much of its range (NSW, QLD and ACT) (*Department of Sustainability, 2013*). Immunological studies are key to understanding the pathogenesis of this disease, assessing the effects of stressors on the immunocompetence of koalas, and assessing responses to vaccines currently under development (*Carey et al., 2010*; *Kollipara et al., 2012*). They are also needed to evaluate the suggestion that Koala Retrovirus (KoRV), is an important agent of immune suppression that may explain koalas' susceptibility to chlamydial infection (*Tarlinton et al., 2005*). KoRV infection occurs at 100% prevalence in northern Australia (New South Wales and Queensland) but has a lower prevalence in the southern states (Victoria and South Australia) (*Simmons et al., 2012*) and, although KoRV has been associated with neoplastic disease in koalas (*Tarlinton et al., 2005*; *Xu et al., 2013*) the relationship between KoRV and infectious diseases is so far unclear (*Simmons, 2011*; *Tarlinton et al., 2005*).

Cytokines that are produced by T helper cells play an essential role in immune defences against intracellular pathogens such as *Chlamydia* (*Landers et al., 1991*; *Perry, Feilzer & Caldwell, 1997*; *Su & Caldwell, 1995*). In humans and mice, resistance to chlamydial infection is associated with Th1-type cytokines, such as interferon $\gamma$ (IFN$\gamma$), which promote cytotoxic T cell responses (*Perry, Feilzer & Caldwell, 1997*). In contrast, a Th2-dominated response, characterised by interleukin 4 (IL-4) and promoted by interleukin 10 (IL-10), is associated with persistence of infection, as these responses inhibit Th1 responses, promote antibody responses that are less able to eliminate intracellular forms of *Chlamydia* and promote structural damage as a result of fibrosis (*Holland et al., 1996*). The similarity of koala chlamydial lesions and serological responses to those of women with *Chlamydia trachomatis* suggests that pathogenesis of the disease may be similar in koalas (*Hemsley & Canfield, 1997*; *Higgins, Hemsley & Canfield, 2005*) but this has been incompletely characterised.

Though no firm evidence exists for an immunomodulatory role of KoRV in koalas, cultured human peripheral blood mononuclear cells (PBMCs) incubated with KoRV increased expression of interleukin 6 (IL-6) and IL-10 (*Fiebig et al., 2006*). IL-10 inhibits the Th1 response by blocking IFN$\gamma$ production (*de Waal Malefyt et al., 1992*) and an excessive IL-10 response is associated with persistence of chlamydial infection in humans

(*Natividad et al., 2008*; *Öhman et al., 2006*). In addition to its role in the differentiation of Th17 cells (*Zhu & Paul, 2008*), IL-6 can modulate the differentiation of naive T cells into Th1 or Th2 cells and shift balance towards a Th2 profile (*Diehl & Rincón, 2002*), and has an important role in increasing inflammation and benefiting chlamydial growth (*Rodriguez et al., 2010*). Study of the role of cytokine expression is well established in immunosuppressive retroviral diseases of humans and cats. The progression of immunosuppressive disease due to HIV in humans is also associated with a shift from Th1 to Th2 cytokine production (*Barcellini et al., 1994*; *Clerici & Shearer, 1993*; *Klein et al., 1997*) and the CD4/CD8 ratio is an established marker for progression of retrovirus induced immune deficiency in HIV-1 positive people (*Margolick et al., 2006*; *Taylor et al., 1989*) and cats infected with feline immunodeficiency virus (FIV) (*Ackley et al., 1990*; *Torten et al., 1991*).

To date it has been impossible to study in depth chlamydial, or potential KoRV, pathogenesis in koalas because of a lack of reagents. Due to pathogen-driven selection, immune molecules are highly divergent, limiting cross-reactivity of available immunological reagents, such as monoclonal antibodies, and impeding development of molecular reagents in the absence of genetic information for the species. Until very recently, studies have been limited to detection of antibodies (*Wilkinson, Barton & Kotlarski, 1995*; *Wilkinson, Kotlarski & Barton, 1994*), lymphoproliferative responses (*Carey et al., 2010*; *Kollipara et al., 2012*; *Wilkinson, Kotlarski & Barton, 1992*), and flow cytometric immunophenotyping of T and B lymphocytes via cross reactive antibodies to CD3, CD79b, MHCII and IFN$\gamma$ at rest and following stimulation with PMA-ionomycin (*Higgins, Hemsley & Canfield, 2004*; *Lau et al., 2012*). Only two studies have quantified expression of koala cytokines by qRT-PCR, detecting up-regulation of IFN$\gamma$, IL-10 and tumour necrosis factor-$\alpha$ (TNF$\alpha$) in response to stimulation with chlamydial antigen (*Mathew et al., 2013a,b*); the range of immune genes that can be studied currently is limited, and the response of koala PBMCs to stimuli used commonly for *in vitro* assessment of cellular immune response in humans and other animals is largely uncharacterised. Further, no reference genes have been validated for use in lymphocyte stimulation assays in koalas. The ability to accurately characterise cytokine expression by qRT-PCR is dependent on the use of validated reference genes, in which the expression does not change with stimulation but it is well known that commonly used reference genes can vary in expression with different mitogen stimulation protocols (*Bas et al., 2004*; *Dheda et al., 2004*; *Glare et al., 2002*).

Here we present a suite of novel koala-specific primers for qRT-PCR of key immune markers of the adaptive immune response; validation of reference genes (GAPDH and 28 s); and their application to quantify expression of the immune genes CD4, CD8$\beta$, IL-10, IL-4, IFN$\gamma$ and IL-6, in koala PBMCs at rest and following three mitogen stimulation protocols: phorbol myristate acetate/ionomycin (PMA-ionomycin), phorbol myristate acetate/phytohemagglutinin (PMA–PHA) and concanavalin A (Con A).

## MATERIALS AND METHODS

### Ethics statement

This study was carried out with the approval of the ethics committee of Taronga Conservation Society Australia/University of Sydney (permit no 4a/01/11) and the New South Wales Government (Scientific licence S10418). All efforts were made to minimise animal suffering. The project was funded by the Australian Research Council Grant LP0989701.

### Sample collection and PBMC stimulation

Study animals were 4 male and 3 female koalas from a captive, chlamydia-free, KoRV-A positive collection, with no history of disease within the past 6 months. Blood (4–8 ml) was collected, in February 2012, from the cephalic vein using manual restraint. It was placed immediately into sodium heparin vials (Vacutainer; Becton Dickinson and Co, NJ, USA) and then transported at room temperature for processing within 3 h of collection. The blood was centrifuged at $800 \times g$ for 10 min, the plasma aspirated, and cells resuspended to 2 times the original blood volume in sterile $1 \times$ PBS. Cells (4 - 5 ml) were overlaid on a double-density gradient consisting of 2 ml Percoll 1.056 g/ml (72.2% PBS, 25% Percoll PLUS, 2.8% 1.5 M NaCl) above 3 ml of Percoll 1.082 g/ml (35% PBS, 58.5% Percoll PLUS, 6.5% 1.5 M NaCl) (GE Healthcare Bio-sciences Corp, NJ, USA), and then centrifuged at $200 \times g$ for 30 min with no brake. PBMCs were aspirated from the interface, transferred to a new tube, and then washed twice with warm PBS supplemented with 0.2% BSA. Cells were then counted on an automated haematology analyser (Sysmex K4500; TOA Medical Electronics Co, Japan), and suspended at $1 \times 10^6$ cells/ml in warm plain media comprised of RPMI 10% FCS (F9423; Sigma-Aldrich, Castle Hill, NSW, Australia). Cell cultures of 1 ml were incubated in plain media (unstimulated) or in media containing PMA 50 ng/ml, Ionomycin 1 $\mu$g/ml (*Higgins, Hemsley & Canfield, 2004*); PMA 25 ng/ml, PHA 2 $\mu$g/ml (*Sullivan et al., 2000*); or Con A 5 $\mu$g/ml (*Lau et al., 2012*) in a sealed tube at 37°C for 5 h. Following incubation, cells were washed twice in warm PBS, resuspended in 1 ml RNAlater (Applied Biosystems, Carlsbad, CA, USA), stored at room temperature overnight and then frozen at −20°C until RNA extraction.

### RNA and cDNA preparation

For RNA extraction, PBMC samples stored in RNAlater (approximately $1 \times 10^6$ cells) were centrifuged at $2300 \times g$ for 15 min. The supernatant of RNAlater was removed, and remaining cells were homogenised and extracted using the RNeasy Mini Kit (Qiagen, Doncaster, VIC, Australia) following manufacturer's instructions. To remove contaminating DNA, RNA preparations were treated with DNase I following manufacturer's instructions (Applied Biosystems, Carslbad, CA, USA). The concentration and purity of RNA was assessed using a NanoDrop ND-1000 Spectrophotometer (Thermo Scientific, Wilmington, DE, USA). RNA integrity was assessed using an Agilent 2100 Bioanalyzer (Agilent Technologies, Waldbronn, Germany). RNA was immediately processed for cDNA synthesis or stored at −80°C.

Peer

Reverse transcription was performed using the Revertaid First Strand cDNA Synthesis kit (Thermo Scientific, Lithuania) according to the manufacturer's instructions, using oligo dT primers. Due to biological variation, 70–700 ng of RNA was used (in 10 $\mu$l) in each reaction. To control for contamination for genomic DNA, 'no reverse transcriptase' (NRT) controls were made using the same protocol omitting reverse transcriptase. cDNA concentration was determined using a NanoDrop ND-1000 spectrophotometer (Thermo Scientific, Wilmington, DE, USA) and then cDNA was stored at −20°C until use in qPCR.

## Generation of partial sequence for koala immune genes

Conserved regions of the genes IL-4, IL-10, IFN$\gamma$, IL-6, CD4, CD8$\beta$ and GAPDH in marsupials, including grey short-tailed opossum *Monodelphis domestica*, tammar wallaby *Macropus eugenii*, Tasmanian devil *Sarcophilus harrisii*, common brushtail possum *Trichosurus vulpecula* (Table 1) were identified using ClustalW (*Thompson, Higgins & Gibson, 1994*) in BioEdit (*Hall, 1999*). PCR amplifications were carried out in an MJ Mini$^{TM}$ thermal cycler (Bio-Rad Laboratories Inc., CA, USA) in 25 $\mu$l reactions containing 0.4 $\mu$M each primer (Sigma-Aldrich, Castle Hill, NSW, Australia), 1× High Fidelity PCR Buffer, 1.5 mM MgSO$_4$, 0.2 mM dNTPs, and 0.75 units of Platinum$^{®}$ Taq DNA Polymerase High Fidelity (Invitrogen, Mulgrave, VIC, Australia). cDNA (10–30 ng) was used for all PCR reactions with the exception of IL-6, where genomic DNA was used as primers designed were positioned within introns due to a scarcity of suitable conserved primer sites within the exons. Cycle conditions for touchdown PCR were: an initial activation step of 95°C for 5 min, then cycles of 94°C for 30 s (denaturation), highest annealing temperature for 30 s (annealing), 72°C for 30 s (extension) with the annealing temperature reducing −1°C per cycle until lowest annealing temperature was reached, followed by 20–25 cycles at the lowest annealing temperature, and a final extension at 72°C for 10 min. Amplicons were visualised by electrophoresis on a 1% agarose gel, and those of expected length were excised, purified using UltraClean GelSpin DNA Extraction Kit (Mo Bio, Calsbad, CA, USA) and sequenced commercially in forward and reverse strands (Macrogen, South Korea). Returned sequence was trimmed to remove low quality sequences and aligned to create a contig using Sequencher$^{®}$ 4.10.1 (Gene Codes, Ann Arbor, MI, USA); and database searches were performed using BLAST (*Altschul et al., 1990*).

## Real-time qPCR of koala immune genes

Partial immune gene sequences obtained (Figs. 1 and 2, Supplementary Material A) were used to design qPCR primers by following recommendations for general primer design for qPCR (*Wang & Seed, 2006*) and using the primer analysis programs Primer 3 (*Rozen & Skaletsky, 2000*) and UNAfold (*Markham & Zuker, 2008*) (Table 1). All qPCR primers designed are located within exons, including IL-6, and annealing temperatures were optimised by gradient PCR. Subsequent qPCR was performed with the following conditions: triplicate qPCR reactions comprised 10 $\mu$l of 2× SsoAdvanced SYBR Green Supermix (Bio-Rad Laboratories Inc., CA, USA), 1 $\mu$l each primer (Table 2), 2 $\mu$l of

**Table 1** Primers developed and used in this study.

| Gene | GenBank/Ensembl sequences | Primers (5′–3′) (q = qPCR primers based on koala sequence) | $T_{Annealing}$ (°C) | Amplicon size (bp) | Proportion of full length coding region amplified (%) |
|---|---|---|---|---|---|
| CD4 | Saha XM_003771252.1 Maeu EF490599.1 Modo NM_001099290.1 | CD4 F: GTGTTCAAGGTGACAGCCA CD4 R:AAGACTCCTGAGCCAGCA qCD4 F: GCCAACCCAAGTGACTCTGT qCD4 R: TCTCCTGGACCACTCCATTC | 56–50<br>60 | 525<br>104 | 40.3 |
| CD8β | Saha XM_003758647.1 Maeu EU152105.1 Modo NM_001146331.1 | CD8 F: ACCCACCAAAAAGACCAC CD8 R: ACCCAGTGGCTTAGGAAAA qCD8 F: ATGTAGCTCGGCACCACTTT qCD8 R: TTGCTGTTCTCATGCCTCAG | 65–50<br>60 | 359<br>159 | 36.2 |
| IL-4 | Saha XP_003756539.1 Maeu ADG01643.1 Modo XP_003339570.1 | IL4 F: GTCTCACATTCCAACTG IL4 R: TGCTCAGGAAATCTTGC qIL4 F: GTTTCCCTGCTTTGAGATGG qIL4 R: TCAGGAAACAGCTTCGGAGT | 56–43<br>55 | 417<br>78 | 89.2 |
| IL-6 | Saha ENSSHAT00000013262 Modo ENSMODT00000001552 | IL6 intron F: GCAAAGGGAAACTCACCA IL6 intron R: AACACCTGTTTGGCTTTTAG<br><br>qIL6 F: TGGATGAGCTGAACTGTACCC qIL6 R: GCTTGCCAAGGATTGTGAGT | 65–50<br><br>60 | 426<br><br>118 | 22.0 |
| IL-10 | Trvu AF026277.1 Saha XM_003767646.1 Modo XM_003340167.1 | IL10 F: ACATGCTCCGAGAWCTTCG IL10 R: TGTAGACTCCTTGTTCC qIL10 F: TTTAGGCGAGAAGCTGAAGG qIL10 R: TCTTCACAGGGCAGGAATCT | 56–43<br>55 | 351<br>69 | 65.2 |
| IFNγ | Saha ENSSHAT00000017850 Maeu ENSMEUT00000007026 Modo AC190119.3 | IFNγ F: CAAGCTNCNTCTTAGCATCC IFNγ R: TGGCTTNTGTTCTGTCTTC<br><br>qIFNγ F: TGAACATGATGGATCGTTGG qIFNγ R: CATTCACTTTGCTGGCAGTG | 65–50<br><br>60 | 466<br><br>186 | 84.3 |
| GAPDH | Saha XM_003771214.1 Modo EF599650.1 | GAPDH F: AGTCCACTGGCGTGTTTACC GAPDH R: GGTCCTCTGTGTAGCCCAAG qGAPDH F: AACTTTGGCATTGTGGAAGGA qGAPDH R: AACATCATCCCTGCTTCCAC | 55<br>55 | 549<br>130 | |

cDNA template, and PCR grade water in a final volume of 20 $\mu$l. Cycling conditions (CFX96 Touch; Bio-Rad Laboratories Inc., CA, USA) were 95°C for 10 min (activation) followed by 40 cycles of 95°C for 15 s (denaturation), and 60°C (55°C for IL-4, IL-10 and GAPDH) for 60 s (annealing and extension). The melting profile was obtained by 90 s pre-melt conditioning at 55°C and then heating the reactions in one-degree increments from 52°C to 95°C with 5 s stops. Primers were titrated to remove primer dimers from no template controls and qPCR products were confirmed by gel electrophoresis, followed by sequencing and alignment, using BLAST searches as previously described. Primers for the candidate reference genes 28 s rRNA (28 s) and koala $\beta$ actin were obtained from the literature (*Daly et al., 2009*; *Markey et al., 2007*).

Efficiency of each reaction, dynamic range, optimal cDNA concentration, $y$-intercept, slope and $r^2$ calibration curve were determined using a triplicate 10× serial dilution of a stimulated sample, covering 8 orders of magnitude. Efficiency was calculated as

**Peer**J

**Table 2  Optimisation values for koala immune gene qPCRs developed in this study.**

| Gene | Efficiency | Conc (nM) | Linear dynamic range (orders of magnitude) | LOQ (copies/$\mu$l) | Cq at LOQ | LOD (copies/$\mu$l) | $r^2$ calibration curve | $y$ intercept | Slope |
|------|-----------|-----------|-------------------------------------------|---------------------|-----------|---------------------|-------------------------|---------------|-------|
| CD4 | 1.02 | 500 | 9 | 4 | 36 | <1 | 0.99 | 43.09 | −3.27 |
| CD8$\beta$ | 1.03 | 500 | 9 | 6 | 36 | <1 | 0.99 | 45.89 | −3.21 |
| IL-4 | 1.05 | 250 | 9 | 10 | 38 | 1 | 0.99 | 47.49 | −3.21 |
| IL-6 | 0.93 | 500 | 6 | 3358 | 36 | <1 | 0.99 | 43.54 | −3.49 |
| IL-10 | 1.06 | 250 | 9 | 0.7 | 36 | <1 | 0.97 | 48.75 | −3.17 |
| IFN$\gamma$ | 0.92 | 300 | 6 | 37 | 33 | 3 | 0.98 | 39.97 | −3.52 |
| GAPDH | 0.97 | 500 | 10 | 3 | 38 | <1 | 0.99 | 42.26 | −3.40 |
| 28S | 0.97 | 250 | 9 | 2 | 35 | <1 | 0.98 | 38.89 | −3.39 |

**Notes.**

Conc, primer concentration (nM); LOQ, limit of quantification; LOD, limit of detection.

$10^{(-1/\text{slope})} - 1$. Standard curves were also constructed using purified PCR product, quantified using a NanoDrop ND-1000 spectrophotometer (Thermo Scientific, Wilmington, DE, USA) and then serially diluted ($1{:}1 \times 10^3$–$1{:}1 \times 10^{12}$) until the curve was no longer linear (limit of quantification; LOQ) and no sample was detected (limit of detection; LOD). DNA concentration was converted to DNA molecule copy number using the concentration and molecular weight of the amplicon (copy number $= [\text{ng DNA} \times 6.022 \times 10^{23}]/[\text{length (bps)} \times 1 \times 10^9\,\text{ng g}^{-1} \times 650\,\text{g mol}^{-1}\,\text{bps}^{-1}]$).

## Expression of cytokine genes by mitogen stimulated PBMC

Samples of the three types of stimulated cells, and the non-stimulated cells, were assayed in triplicate using each set of cytokine primers and the potential reference gene primers. cDNA (200 ng in 2 $\mu$l) was used in each reaction, based on the calculated dynamic range of the assay. No template (NT) and no reverse transcriptase (NRT) control samples were run in triplicate with each primer set.

Raw Cq values were transformed to $2^{-\text{Cq}}$ for analysis. The $2^{-\Delta\text{Cq}}$ method was used to assess difference between unstimulated samples as well as between mitogen stimulation protocols (*Schmittgen & Livak, 2008*) and to calculate reference gene relative expression (*Dheda et al., 2004*). The $2^{-\Delta\Delta\text{Cq}}$ method was used to assess cytokine expression fold change (*Schmittgen & Livak, 2008*). Normality was assessed using a Shapiro–Wilk test (GraphPad Prism 6.03; GraphPad Software Inc., CA, USA). As data were not normally distributed, Mann–Whitney U test and Freidman's tests were performed, using Minitab 16 (Minitab Inc., PA, USA). A $P$ value of <0.05 was considered significant.

## RESULTS

### Partial sequence of koala immune genes

Partial koala sequences obtained and used for primer design showed >58% amino acid homology with relevant genes of opossum species, and >71% and 78% with available tammar wallaby and Tasmanian devil amino acid sequences, respectively.

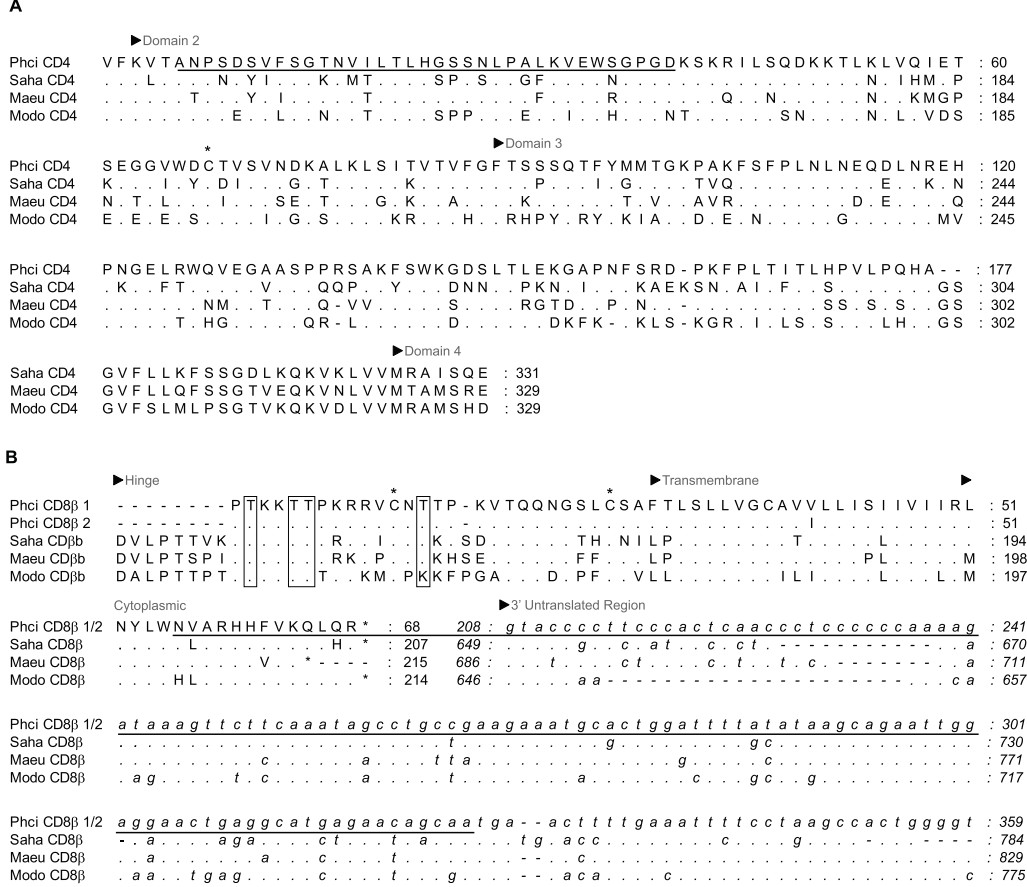

**Figure 1** **Alignment of deduced amino acid sequence of koala (A) CD4 and (B) CD8β with other marsupials.** Dashes indicate gaps introduced to optimise the alignment, and dots indicate identity with the koala. Koala qPCR targets are underlined. * indicates extracellular cystine residue in CD4 and CD8β, and open box in CD8β indicates potential N-linked and O-linked glycosylation sites. GenBank accession numbers from grey short-tailed opossum (Modo), tammar wallaby (Maeu) and Tasmanian devil (Saha) are listed in Table 1.

GenBank/Ensembl sequences for alignment, forward and reverse primers, optimum annealing temperature and amplified fragment size are all shown in Table 1. All sequences possessed key conserved residues, confirming their identity (Figs. 1 and 2, Supplementary Material A). Sequences for IL-10, and IFNγ were consistent with those of *Mathew et al.*, (*2013a*; *2013b*). Sequences have been lodged in GenBank under the accession numbers: PhciCD4, KF731654; PhciCD8b*01, KF731655; PhciCD8b*02, KF731656; PhciIFNγ, KF731657; PhciIL4, KF731658; PhciIL10, KF731659; PhciGAPDH, KF731660; and PhciIL6, KF731661.

## qPCR optimization and validation

All assays were highly sensitive, had a linear dynamic range greater than 6 orders of magnitude and had an optimal amplification efficiency of 92% or greater (Table 2). The

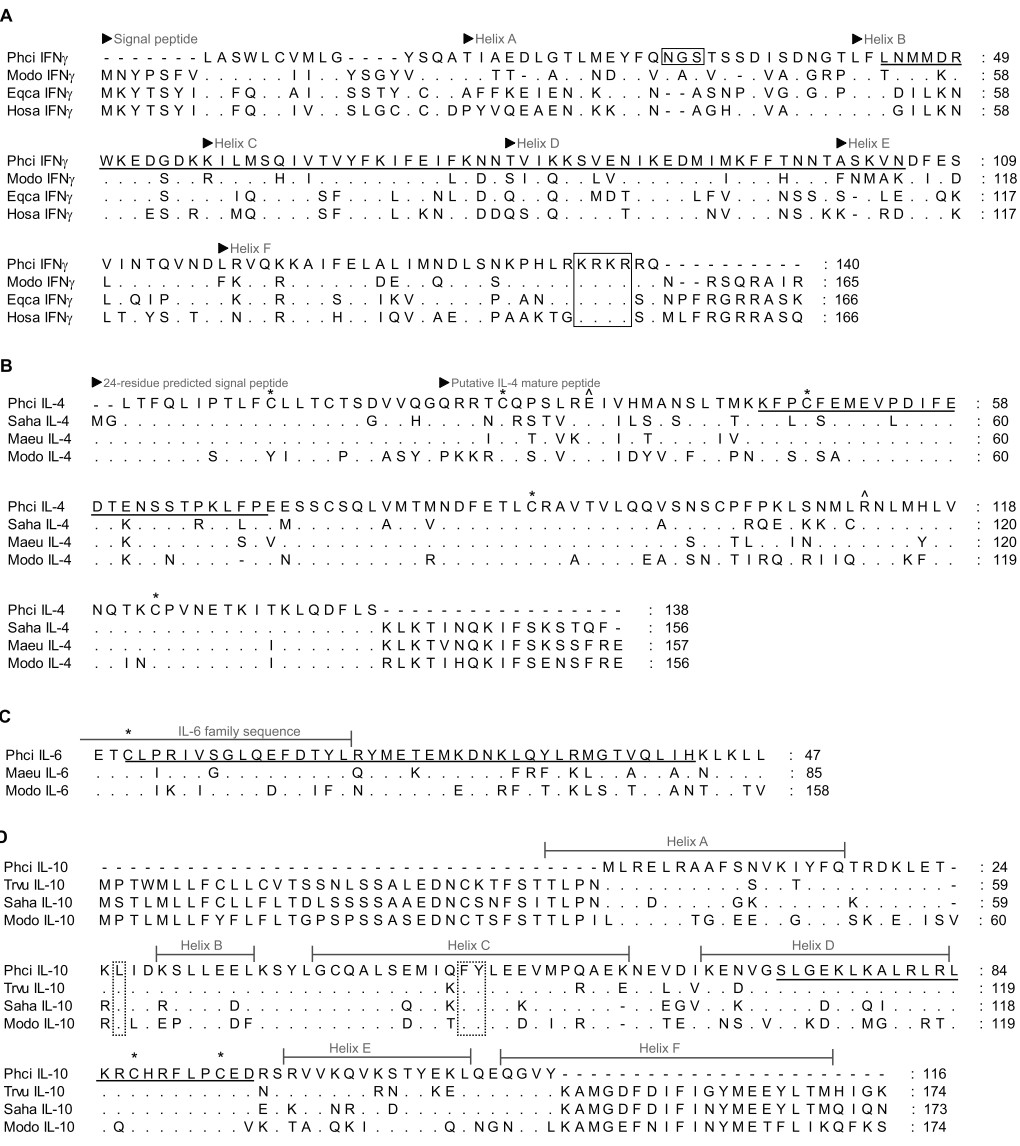

**Figure 2 Alignment of deduced amino acid sequence of koala (A) IFNγ, (B) IL-4, (C) IL-6 and (D) IL-10 with other marsupials and mammals.** Dashes indicate gaps introduced to optimise the alignment, and dots indicate identity with the koala. Koala qPCR targets are underlined. * indicates conserved cysteine residues that form essential disulphide bonds in human IL-4, IL-6 and IL-10. ^ indicates conserved residue for high affinity receptor interaction in human IL-4. Open box in IFNγ indicates potential N-linked glycosylation site and conserved NLS (KRKR). Dotted box in IL-10 indicates residues for structural stabilisation. GenBank accession numbers are listed in Table 1 with the addition of eutherians due to limited marsupial sequence available (American mink (Nevi) IL-6 P41693.1, horse (Eqca) IFNγ ABS28998.1 and human (Hosa) IFNγ CAA26022.1).

relative expression results for reference genes are shown in Fig. 3. Of the three candidate reference genes, 28 s and GAPDH were considered suitable for all treatments, with a mean relative expression usually <2, and a maximum relative expression <3.5 (*Dheda*

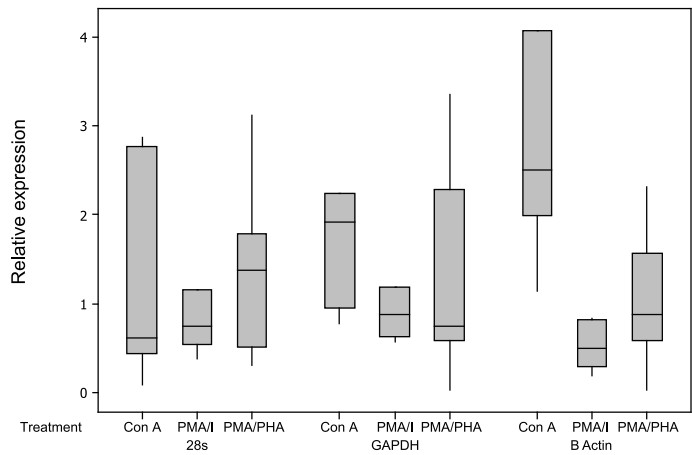

**Figure 3** **Relative expression of potential reference genes following mitogen stimulation of koala lymphocytes.** Relative expression was calculated using $2^{-\Delta Cq}$ using the unstimulated sample as the control, i.e., $2^{-(\text{Cq of mitogen stimulated sample} - \text{Cq of unstimulated sample})}$. Box plots of Median, 1st and 3rd quartiles (boxes), and $1.5\times$ quartiles (whiskers). Asterisk indicates statistically significant up or down regulation of expression ($*P$ value $<0.05$, $**P$ value $<0.01$). PMA/I = phorbol myristate acetate/ionomycin, PMA/PHA = phorbol myristate acetate/phytohemagglutinin, Con A = concanavalin A.

*et al., 2004*). $\beta$ actin expression was significantly up-regulated by Con A and was significantly down-regulated by PMA-ionomycin treatment, and was therefore not considered a suitable reference gene for this experiment, although it appears appropriate for use in experiments using PMA–PHA stimulation. Results for all cytokines and all mitogen stimulation protocols were highly similar when calculated as $2^{-Cq}$ values (not-normalised) or $2^{-\Delta Cq}$ values (normalised against the geometric mean of the two reference genes, 28 s and GAPDH) further supporting suitability of GAPDH and 28 s as reference genes (*Schmittgen & Livak, 2008*).

## Cytokine and CD4/CD8$\beta$ expression in unstimulated samples

The $2^{-\Delta Cq}$ cytokine expression in un-stimulated samples varied markedly among individuals, the median $\times 10^{-4}$ (min − max)$2^{-\Delta Cq}$ values are shown in Fig. 4. The expression levels of CD4 and CD8$\beta$ are much higher than the cytokine levels. The individual with the highest IL-4 and IL-10 expression also had low IFN$\gamma$ expression ($0.9 \times 10^{-4}$), consistent with a Th2 dominated profile. The CD4/CD8$\beta$ expression ratios for the unstimulated samples varied from 0.1 to 6.3 with a median value of 2.1.

## Response of PBMC to mitogen stimulation

Based on $\Delta\Delta Cq$ results (Fig. 5), the most potent up-regulation of IFN$\gamma$ (median fold-change 523) and IL-4 (median fold-change 96) was induced by PMA-ionomycin, though all protocols increased expression of IFN$\gamma$. IL-4 was also induced by Con A (median fold-change 5) but not PMA–PHA. IL-10 was also induced but less dramatically; PMA-ionomycin induced a median fold-change of 4.0, while PMA–PHA and Con A did not measurably increase its expression. There was no significant change in IL-6

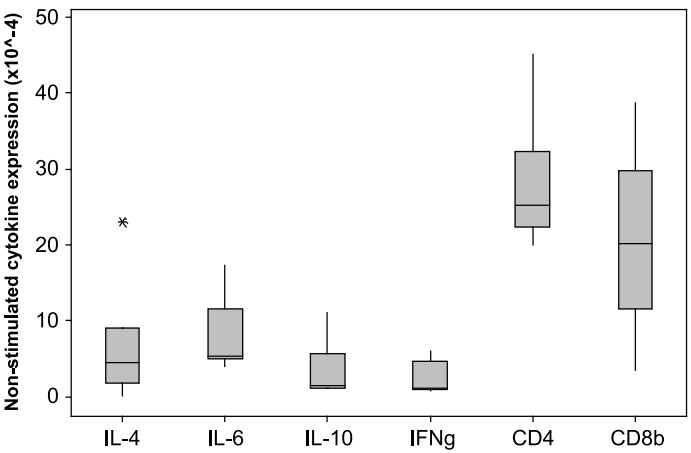

**Figure 4 Cytokine and CD gene expression in unstimulated koala lymphocytes.** Cytokine gene expression was calculated using $2^{-\Delta Cq}$ where the internal control is the geometric mean of the 2 reference genes 28 s and GAPDH, i.e., $2^{-(Cq\ of\ gene\ of\ interest - Cq\ of\ internal\ control)}$. A single outlying value (CD4, 245) has been omitted for scale. Box plots of Median, 1st and 3rd quartiles (boxes), and 1.5× quartiles (whiskers).

expression overall with any protocol, though two individuals exhibited 2.8 and 5.3 fold changes in response to PMA–PHA. Expression of CD4 was significantly down-regulated with PMA–PHA stimulation and CD8$\beta$ was down-regulated with all mitogen stimulation protocols. There was no apparent sex difference in expression of any of the cytokines in the unstimulated or mitogen-stimulated samples, though the sample size was too small for statistical analysis (data not shown). Individual responses to PMA-ionomycin stimulation varied markedly, ranging from fold-changes of 11 to 189 for IL-4, 0.3 to 7 for IL-10 and 25 to 2985 for IFN$\gamma$.

## DISCUSSION

This study reports the first partial cDNA sequence for koala CD4, CD8, IL-4, and IL-6 and describes changes to cytokine and CD4 and CD8$\beta$ expression in koala PBMCs following stimulation by three commonly used mitogens, using qPCR and validated reference genes. While expression profiles of CD4, CD8$\beta$, IFN$\gamma$, IL-4 and IL-10 were similar to those of other species, those of IL-6 gave unexpected results.

In addition to facilitating development of the qPCR assays in the present study, the partial cDNA sequences obtained in this study will be of use in development of additional immunological reagents for the koala, including antibodies for use in flow cytometry. While qPCR is able to give an overview of cytokine production, flow cytometry is important for evaluating the activity of particular cell subsets. No antibodies exist to label koala CD4, and all cross-reactive antibodies currently in use for labelling other CD molecules of koalas label conserved intracellular epitopes, requiring fixation and permeabilisation of cells before labelling. The CD4 sequence described here covers

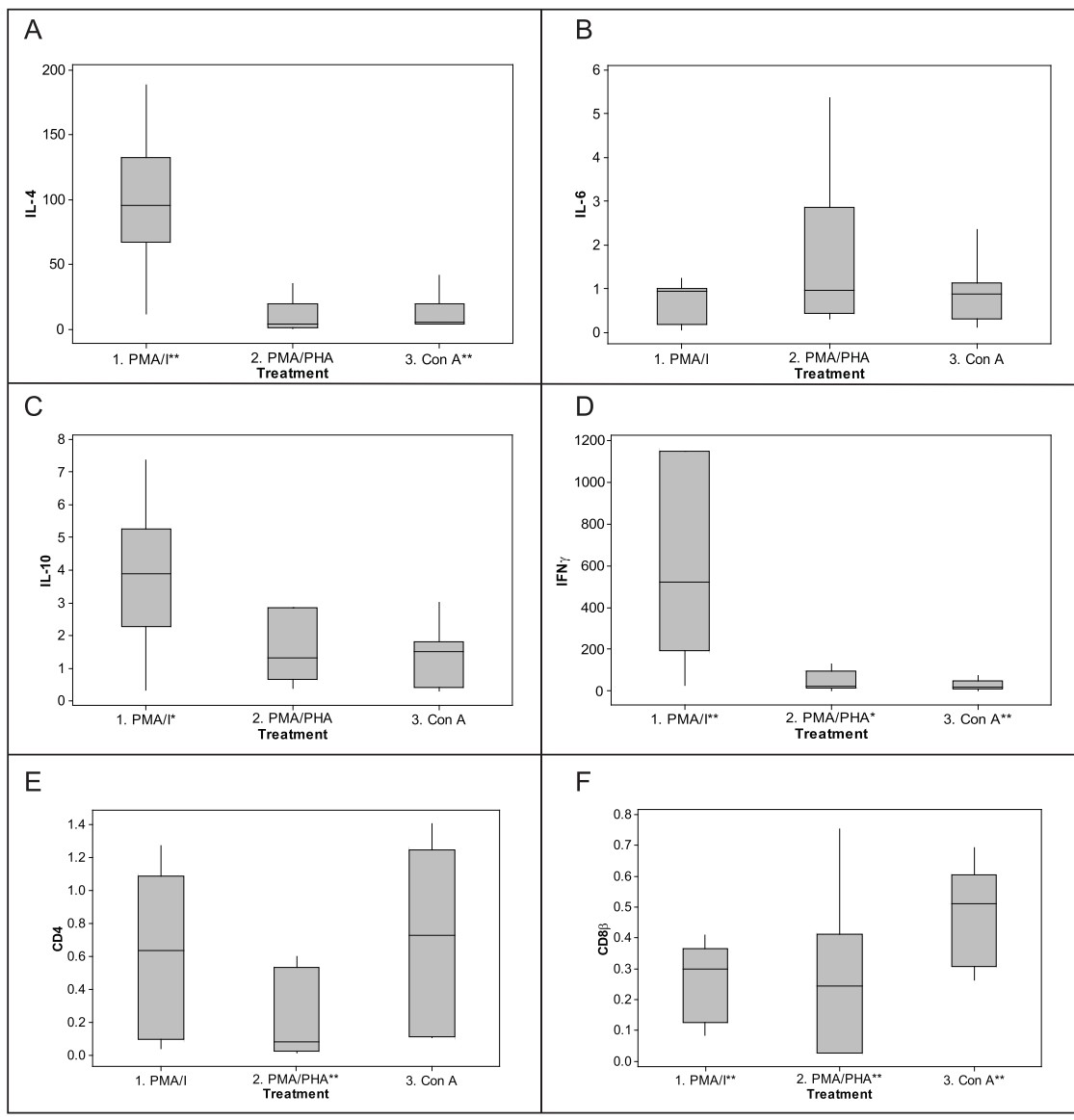

**Figure 5** **Fold-change response to mitogen stimulation by cytokine and CD genes in koala lymphocytes. (A) IL-4, (B) IL-6, (C) IL-10, (D) IFNγ, (E) CD4, (F) CD8β.** Fold change was calculated using the $2^{-\Delta\Delta Cq}$ method, i.e., $2^{-[[(Cq \text{ of gene of interest}-Cq \text{ internal control}) \text{ mitogen stimulated sample}]-[(Cq \text{ of gene of interest}-Cq \text{ internalcontrol}) \text{ unstimulated sample}]]}$. The internal control is the geometric mean of the two reference genes, 28 s and GAPDH. Box plots of Median, 1st and 3rd quartiles (boxes), and 1.5× quartiles (whiskers). Asterisk indicates statistically significant up or down regulation of expression (*P value < 0.05, **P value < 0.01). PMA/I = phorbol myristate acetate/ionomycin, PMA/PHA = phorbol myristate acetate/phytohemagglutinin, Con A = concanavalin A, IL-4 = interleukin 4, IL-6 = interleukin 6, IL-10 = interleukin 10, IFNγ = interferonγ.

external domains, making it applicable to development of antibodies for labelling and isolating CD4$^+$ cells for manipulation of this important group of lymphocytes.

The stimulation protocols in this study used known T cell mitogens, covering two stimulatory pathways. PHA is the most commonly used lectin to stimulate human T cells

and Con A is considered the most effective stimulant for mouse T cells (*Hodgkin, 2001*). Both employ surface molecules for cell activation, whereas PMA-ionomycin triggers pathways intracellularly. Down-regulation of CD8$\beta$ (all mitogen protocols) and CD4 (PMA–PHA) were expected findings as, in humans, mitogen stimulation with PMA-ionomycin and PHA leads to rapid down-regulation of membrane expression of CD4 (*Baran et al., 2001*; *Petersen et al., 1992*) and a similar pattern may exist for CD8$\beta$. All three mitogens induced significant up-regulation of IFN$\gamma$, while PMA-ionomycin and Con A both up-regulated IL-4. As in other species (*Baran et al., 2001*; *Gonzalez et al., 1994*; *Rostaing et al., 1999*), and consistent with previously described IFN$\gamma$ responses in koala and brushtail possum PBMCs (*Higgins, Hemsley & Canfield, 2004*), PMA-ionomycin elicited very strong up-regulation of IFN$\gamma$ and IL-4; whereas the up-regulation of IFN$\gamma$ and IL-4 by PMA–PHA and Con A was more modest, but much less variable, which may be useful in some applications.

Although the short incubation times employed in the current study would be an advantage when stimulating samples from this wildlife species in the field, it may be that the study of IL-6 and IL-10 will require more extended incubation and, possibly, specific mitogens. Up-regulation of IL-10 in the present study was less than that reported by *Mathew et al. (2013a)* following antigen stimulation (by UV inactivated *C. pecorum* particles). In addition, although some individuals showed evidence of IL-6 up-regulation, there was no significant increase in IL-6 expression among treatment groups. Artificially low qPCR results can be obtained where RNA degradation prevents first strand cDNA synthesis, especially when oligodT primers are used and when targets are greater distances from the 3' poly-A tail. However, we consider this unlikely in the present study as RNA quality was assessed as high according to recent MIQE guidelines (*Bustin et al., 2009*) (data not shown), and IL-10 and IL-6 primers were no further from the 3' end than the other primers in the present study. Inappropriate reference genes can also affect results, but the reference genes GAPDH and 28 s were validated in the current study, based on degree of fold-change, along with comparison of transformed Cq values to $\Delta$Cq values across treatment groups. We consider it most likely that further, cytokine-specific, optimization of stimulation protocols will increase up-regulation of these cytokines. The greater up-regulation of IL-10 in *Mathew et al. (2013a)* was observed after 12 h of stimulation and production of IL-6 and IL-10 has been shown to continue to increase up to 12 h and beyond in other studies (as opposed to the 5 h of stimulation employed in the current study) (*Baran et al., 2001*; *Fagiolo et al., 1993*; *Liu et al., 2009*). In addition to increasing the period of incubation, specific mitogens may be ideal; while PMA-ionomycin was the most potent stimulant of IFN$\gamma$ and IL-4 responses, PHA-PMA showed the most promise for up-regulation of IL-6, consistent with *Fagiolo et al. (1993)*.

The assays developed in the present study were sufficiently sensitive to quantify cytokine expression in resting koala lymphocytes, and detection of a pattern of expression consistent with a Th2-dominated response in one individual in the study suggests that the assays may be useful for evaluation of *in vivo* immune responses in naturally occurring disease. In addition to the benefits of directly observing natural processes, the ability to

observe immune responses without the need for lymphocyte culture in the field is an advantage in this free-ranging species.

This study provides valuable sequence information and validated assays for evaluation of the adaptive immune response of the koala. These assays, and the fundamental information on koala lymphocyte responses obtained with them, pave the way for new understanding of koala's susceptibility to disease, and how it might be influenced, for better or worse, by vaccination, environmental stressors, or retroviral infection.

## ACKNOWLEDGEMENTS

We thank Lisa Black and the staff of Taronga Zoo for assisting with blood sample collection, and Associate Professor Peter Williamson for access to laboratory equipment.

### Funding

This work was funded by an Australian Research Council Linkage Project grant, in collaboration with the University of Queensland and supported by Australia Zoo/Wildlife Warriors Worldwide Ltd and Queensland Parks and Wildlife Service as partners The funders had no role in study design, data collection and analysis, decision to publish, or preparation of the manuscript.

### Grant Disclosures

The following grant information was disclosed by the authors:
Australian Research Council LP0989701.

### Competing Interests

The authors declare that they have no competing interests.

### Author Contributions

- Iona E Maher and Quintin Lau conceived and designed the experiments, performed the experiments, analyzed the data, wrote the paper, prepared figures and/or tables, reviewed drafts of the paper.
- Joanna E Griffith conceived and designed the experiments, performed the experiments, analyzed the data, wrote the paper, reviewed drafts of the paper.
- Thomas J Reeves performed the experiments, analyzed the data, reviewed drafts of the paper.
- Damien P Higgins conceived and designed the experiments, performed the experiments, analyzed the data, contributed reagents/materials/analysis tools, wrote the paper, prepared figures and/or tables, reviewed drafts of the paper.

### Animal Ethics

The following information was supplied relating to ethical approvals (i.e., approving body and any reference numbers): Animal Ethics Committee of Taronga Conservation Society Australia/University of Sydney (permit no 4a/01/11).

## DNA Deposition

The following information was supplied regarding the deposition of DNA sequences:

Genbank

PhciCD4, KF731654;

PhciCD8b*01, KF731655;

PhciCD8b*02, KF731656;

PhciIFN, KF731657;

PhciIL4, KF731658;

PhciIL10, KF731659;

PhciGAPDH, KF731660;

PhciIL6, KF731661

## Supplemental Information

Supplemental information for this article can be found online at
http://dx.doi.org/10.7717/peerj.280.

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
