# Peer review of "Expression profiles of the immune genes CD4, CD8β, IFNγ, IL-4, IL-6 and IL-10 in mitogen-stimulated koala lymphocytes (Phascolarctos cinereus) by qRT-PCR"

_PeerJ, doi:10.7717/peerj.280_

## Round 0.1 · original submission · Minor Revisions

The two reviewers of your manuscript had a number of concerns that should be addressed.

Reviewer 1 ·

Basic reporting

see comments below

Experimental design

acceptable

Validity of the findings

see comments below

Additional comments

The manuscript by Maher et al describes the sequences of immune genes CD4, CD8b, IFNg, IL4, IL6 and IL10 from the koala and their expression in unsimulated and stimulated koala PBMCs. The development of qPCR assays for the koala is an important step in characterizing the immune response of the koala in health and in disease in this important species for which species specific reagents are currently lacking. The significance of the paper is diminished somewhat due to the recent publication of two other papers describing qPCR assays of koala cytokines IFNg and IL10 (Mathew et al). A recent publication on koala IL6 has also now been published which is not cited by the authors (Cunningham et al., 2013). Nevertheless, this paper still provides the first description of qPCR assays for IL4, CD4 and CD8 and this aspect could be emphasized to a greater extent. However, the manuscript could be improved before it is accepted for publication.

Specific comments:

Abstract:
What are “developmental vaccines”?
Change “common opossum” to South American opossum

Introduction
Line 32: replace ; with a . after the Tarlinton et al reference
Line65-69: indicate that these reagents were cross-reactive reagents and not species specific.

Materials and methods:
Primer design: this section is far more than simply primer design. It includes PCR, sequencing, sequence analysis and qPCR. This section should be split up into sections or described more appropriately.
Line 132: “common opossum” should be changed to the South American opossum.
Line 139-140: why was there a need to position primers within the IL6 intron?

Results:
Partial sequences were obtained for the koala immune genes. Although a table proves information on amplicon length etc, this section could more adequately describe the amount of sequence obtained relative to the full length genes. More description of the sequences and specific details on the conserved residues present in each sequence could also be provided.

qPCR optimization and validation
The first few sentences of this section read more like a shopping list than a description of the results.

Cytokine and CD4/CD8b expression in unstimulated samples
Indicate the number of individuals included in each analysis.

Discussion
Line 253: the sentence” …ConA is considered the most effective stimulant for mouse cells”: insert T cells here.

Line 268: What was the antigen used for stimulation?



Figure legend 1:
Indicate what the species are that are abbreviated in the alignment and accession numbers similar to figure legend 2. Why are eutherian mammals included in figure 2 but not figure 1?
Figure legend 2:

Reviewer 2 ·

Basic reporting

Abstract: The term ‘T suppressor’ should be replaced by ‘T regulatory’. ‘T suppressor’ is less commonly used in current immunology than the widely accepted ‘T regulatory’, possibly because historically, ‘T suppressor’ is associated to the period of ‘dark ages’ in immunology, and also refers to the function rather than a particular subset of CD4+ T cells. On the other hand, T regulatory (Treg) cells have been identified as Foxp3+CD25+CD4+ T cells. Similarly, Th1 cells are generally Tbet+CD4+; Th2 cells are generally Gata3+CD4+.

Introduction: Lines 53-54. While the authors cited Diehl & Rincon 2002, to support the notion that IL-6 skews naïve CD4+ T cells to Th2, it is worth noting that this is uncommon in the current Th dogma. IL-6 or IL-21, in combination with TGFb, gives rise to Th17 cells, whereas IL-4, IL-2, and others are critical for Th2 differentiation (See reviews from William E Paul).

Lines 48-60. This paragraph highlights why delineating Th responses are important. It will be less confusing for a reader if the authors can introduce a line or two before shifting into the evidence from HIV in humans.

Cytokine and CD4/CD8b expression in unstimulated samples: Line 214 to 220. The reader will benefit if the results reported in this paragraph are plotted on a graph and shown as a figure.

Line 230 to 232: Data not shown should be included at the end of the text stating ‘there was no apparent sex difference in expression of any of the cytokines …’.

Discussion: Line 272 and 273. Data for the RNA quality should either be shown or at least stated as ‘data not shown’ since the authors have assessed the RNA quality using the Bioanalyzer.

Acknowledgements: Perhaps the source of funding should be included.

Experimental design

IL-6 primer design: It is noted in the section ‘Primer design’ that the IL-6 primers sit in the introns and genomic DNA was used for sequencing. Could the authors clarify that the subsequent qPCR design for IL-6 primers is in the exons?

Validity of the findings

Calculation of Cq values: The authors stated that 2^-delta(d)Cq was used to calculate reference gene fold change, whereas 2^-ddCq was used to assess cytokine expression fold change. They should make a distinction between 2^-dCq and 2^-ddCq. The formula of 2^-dCq is 2^-((Cq of gene of interest) - (Cq of reference gene)), which gives relative expression of the gene of interest normalised to reference genes. In contrast, the formula of 2^ddCq is 2^-((dCq of 1 test sample) - (dCq of 1 control sample)), which gives the differential expression of 1 gene in the test sample relative to the control. Hence, 2^-ddCq is referred to as fold change, whereas 2^-dCq should be referred to as relative expression.

Non-Gaussian assumption: The authors noted that their data did not follow a normal distribution without indicating how they arrived at this notion. A normality test (i.e. D'Agostino-Pearson) using Prism (GraphPad) will suffice.

Figure 3: The y-axis and legend should read ‘relative expression’, since 2^-dCq was used for calculation. Because Figure 3 shows the results of reference genes, the authors should indicate how they derived 2^-dCq. For example, gene 28S in sample ConA, what Cq was used for the subtraction from Cq of gene 28S? Since the same amount of cDNA is loaded in each qPCR reaction, the lower the standard deviation of Cq or dCq for the reference gene across all samples will indicate that the gene is more likely to be expressed stably and suitable as a reference gene. A table showing the standard deviation for the 3 reference genes will be helpful (See Silver N, et al. 2006 7:33, BMC Molecular Biology, Vandesompele J, et al. 2002 3: research0034, Genome Biology). That said, the results shown in Figure 3 are valid and in line with the authors’ conclusion and subsequent findings.

Figure 4: This figure shows the fold change (differential expression) of the different cytokine genes. However, the authors did not indicate which control sample was used for comparison. In these figures, the control sample should be 1, since fold change = 2^-((dCq of control sample)-(dCq of control sample)) = 1.

Additional comments

In this report, Maher I. et al. developed a molecular quantitative approach to identify immune responses in koalas. This technique, combined with previous work using flow cytometry from this group, will set the stage for examining the cytokine responses of specific immune cells in diseased koalas in future studies.

---

## Round 0.2 · accepted · Accept

Your paper has been accepted for publication.